# Nuclear Magnetic Resonance (NMR) Metabolomics: Current Applications in Equine Health Assessment

**DOI:** 10.3390/metabo14050269

**Published:** 2024-05-07

**Authors:** Fulvio Laus, Marilena Bazzano, Andrea Spaterna, Luca Laghi, Andrea Marchegiani

**Affiliations:** 1School of Biosciences and Veterinary Medicine, University of Camerino, Via Circonvallazione, 93/95, 62024 Matelica, Italy; fulvio.laus@unicam.it (F.L.); andrea.spaterna@unicam.it (A.S.); andrea.marchegiani@unicam.it (A.M.); 2Department of Agricultural and Food Sciences, University of Bologna, 47521 Cesena, Italy; l.laghi@unibo.it

**Keywords:** horse, equine, metabolomics, NMR, diagnostic markers, diseases, donkey, metabolites

## Abstract

Metabolomics can allow for the comprehensive identification of metabolites within biological systems, at given time points, in physiological and pathological conditions. In the last few years, metabolomic analysis has gained popularity both in human and in veterinary medicine, showing great potential for novel applications in clinical activity. The aim of applying metabolomics in clinical practice is understanding the mechanisms underlying pathological conditions and the influence of certain stimuli (i.e., drugs, nutrition, exercise) on body systems, in the attempt of identifying biomarkers that can help in the diagnosis of diseases. Proton Nuclear Magnetic Resonance spectroscopy (^1^H-NMR) is well tailored to be used as an analytical platform for metabolites’ detection at the base of metabolomics studies, due to minimal sample preparation and high reproducibility. In this mini-review article, the scientific production of NMR metabolomic applications to equine medicine is examined. The research works are very different in methodology and difficult to compare. Studies are mainly focused on exercise, reproduction, and nutrition, other than respiratory and musculoskeletal diseases. The available information on this topic is still scant, but a greater collection of data could allow researchers to define new reliable markers to be used in clinical practice for diagnostic and therapeutical purposes.

## 1. Introduction

Metabolomics can be defined as the global assessment of metabolites present in a biological system in each physiological state at a given time point [1,2,3]. In the scientific literature, the terms metabolomics and metabonomics are used interchangeably, although the latter defines the assessment in response to biological stimuli (e.g., diseases, nutritional supplementation, exercise) or genetic manipulation [2].

Metabolomics approaches have been applied in several medical research areas with a variety of purposes, including identifying diseases’ pathways and related diagnostic markers. In the last few years, metabolomics has gained popularity, both in human and in veterinary medicine, showing great potential for novel applications in clinical activity [4,5].

Metabolomic studies can be performed through either a targeted or an untargeted approach [6]. Targeted analysis identifies and quantifies specific metabolites, selected on the basis of previous knowledge. Untargeted approaches aim to comprehensively capture information across the metabolome, subsequently endeavoring to pinpoint biomarkers of interest, without relying on previous knowledge. For instance, this may involve distinguishing between case and control groups [6,7].

The most common analytical platforms used for metabolomic analyses are Mass Spectrometry (MS) and proton Nuclear Magnetic Resonance (^1^H-NMR) spectroscopy [3]. MS is a very sensitive analytical platform that can be coupled with chromatographic techniques such as Liquid Chromatography (LC/MS) or Gas Chromatography (GC/MS) for molecules’ separation prior to identification [3,8].

^1^H-NMR is an analytical technique with a high start-up cost and a sensitivity far lower than LC/MS or GC/MS. These limitations are counterbalanced by the possibility of application to biomaterials after minimal preparation, often limited to pH adjustment and centrifugation. Moreover, it allows for the simultaneous absolute quantification of molecules of biological interest with very different functional groups [3,8,9,10,11,12]. The only factors that modulate an ^1^H-NMR spectrum are the solvent, the magnetic field, and the pulse sequence employed to transfer magnetization to the protons [13]. This leads to minimal reliance on analysis conditions, facilitating reproducibility. Figure 1 offers a visual impression of the extent of this peculiarity. It shows a superimposition of the spectra that some of the authors of the present review registered on the synovial fluids of horses’ joints, healthy or affected by osteoarthritis [4]. Even if the only post-processing steps of the spectra were referencing alanine’s signals at 1.46 ppm and normalization, the horizontal position of the signals (chemical-shift) from branched amino acids appears as perfectly consistent across the entire samples’ set.

The investigation about possible markers that can be used for diagnostic and therapeutic purposes is very active in human medicine, and this is demonstrated by the publication of several recent papers concerning this topic [14,15,16]. Metabolomics can be used to clarify the pathogenetic mechanisms underlying certain diseases or even for research concerning mechanisms underlying the interaction between pathogens and the host. In this context, its potential for future application is largely recognized [17].

In veterinary medicine, metabolomic research is still limited, although its potential, as previously stated, is widely recognized [18,19].

In this mini-review article, we examined publications focusing on the use of ^1^H-NMR to equine biological fluids ex vivo, with the aim of investigating the current availability of validated diagnostic markers and future perspectives of this analysis in equine veterinary clinical practice.

## 2. Methods

In February 2024, we referred to the following databases: PubMed, Web of Science, Google scholar, and Wiley Online Library. The search terms included the following: Equine(s), horse(s), equid(s), “Equine NMR”, “Equine Metabolomics”, “Equine metabolomic marker(s)”, and “Equine metabonomic”. Only journal articles in English were used. No time period was specified, and duplicate articles were removed.

## 3. Results

Research on equine metabolomics using ^1^H-NMR as a platform can be categorized into five main areas, such as: (i) the respiratory system; (ii) musculoskeletal system; (iii) exercise; (iv) reproduction; (v) nutrition/production. NMR metabolomic assays have been applied to several biological fluids, with plasma being the mostly observed matrix. Figure 2 shows the different samples analyzed in different papers.

### 3.1. Metabolomics in Respiratory Diseases

While there is a plethora of research focusing on metabolomics of respiratory samples performed by ^1^H-NMR in humans [20,21], only two works have been found on equine species (Table 1).

We previously set up research on horses affected by asthma, sampling and analyzing bronchoalveolar lavage fluid (BALF), tracheal wash (TW), and exhaled breath condensate (EBC) [22,23].

The results of this research showed that the metabolomic analysis of BALF, TW, and, in some ways, EBC, has the potential to serve as a diagnostic tool in horses with asthma since higher concentrations of histamine and oxidant agents, (i.e., glutamate, valine, leucine, and isoleucine), as well as lower levels of ascorbate, methylamine, dimethylamine, and O-phosphocholine, were found in the TW of sick horses compared to the control group. Regarding BALF, asthmatic horses showed higher levels of formate and isopropanol and lower levels of myo-inositol and glycerol [22]. Among these, the most significant metabolite is myo-inositol since it is involved in the normal physiology of the lungs [24,25,26]. Research in humans demonstrated that its decrease in plasma is associated with a higher risk of respiratory distress syndrome [27]. Future research studies are needed to possibly confirm the use of this metabolite as a diagnostic marker of respiratory disease in equines. No differences were found for EBC between sick and healthy horses in one study [23], while a significant increase in the ethanol and methanol concentration resulted in being related to the presence of pulmonary disease in another research study [22]. Compared to healthy subjects, ethanol was also found to be higher in the EBC of human patients suffering from pulmonary cystic fibrosis [28].

The metabolomic approach in human respiratory diseases is advancing quite quickly, while it has just started in equine medicine. The use of biofluids of respiratory origin such as EBC has the advantage of being accessible through non-invasive or minimally invasive procedures, without sedation and adverse effects. The implementation of studies on EBC may facilitate the characterization and follow-up of asthma in horses, in comparison to classical techniques (e.g., bronchoalveolar lavage). Given its potential uses, especially for the non-invasive methodology of samples collection, it is warmly desirable that further clinical research be performed to better clarify the potential of ^1^H-NMR in respiratory disease.

### 3.2. Metabolomics in the Musculoskeletal System

Regarding the musculoskeletal system, five publications applied ^1^H-NMR to investigate osteoarticular diseases by sampling synovial fluid (SF) or plasma [4,29,30,31,32] (Table 2).

In our previous study, phenylalanine showed a marked decrease in horses affected by osteoarthritis (OA) [4], in agreement with the result obtained in humans affected by rheumatoid arthritis [33]. An interesting work by Anderson et al. describes a metabolomic analysis of SF collected from different horses’ joints affected by OA (*n* = 4), osteochondrosis (*n* = 6), and synovial sepsis (*n* = 7) [30]. The authors found higher concentrations of phenylalanine in the SF from nonseptic compared to septic joints, in agreement with our results. Phenylalanine is an essential amino acid that can be converted into L-DOPA, thyroxine, adrenaline, and norepinephrine, and it can be also involved in the regulation of several enzymatic activities. Although its involvement in joint diseases has been established, the exact role of phenylalanine in joint diseases and the meaning of its variation in horses needs to be clarified.

Glutamine was found in a lower concentration in the SF of horses affected by osteoarthritis [4]. In 2017, a study conducted by Takahashi et al. showed a higher consumption of glutamine in rats with inflammatory joint diseases [34]. These findings are in line with our results. Furthermore, glutamine seems to further increase in subjects with septic joint disease [30], which makes it an interesting metabolite for future diagnostic/therapeutic uses, as well as for understanding the pathogenesis of this disease.

Metabolites primarily associated with energetic metabolism exhibited inconsistencies across different studies. While glucose and lactate seem to increase in both septic and nonseptic joint forms [29,30] or, at best, they exhibit stability [4], opposite results were obtained for creatine and creatinine, which showed different increase/decrease fluctuations [4,29,30].

^1^H-NMR metabolomics applied to plasma after laminitis induction with oligofructose revealed an increase in some energy-linked metabolites (e.g., lactate and 3-hydroxy-butyrate) [31]. To date, this is the only study on the application of ^1^H-NMR in horses affected by laminitis, and its results should be considered cautiously. Low-density lipoproteins (LDLs) and phosphatidylcholine concentrations have been found to increase in horses after the induction of laminitis, probably due to the modulation of liver fat metabolism or damage to intestinal mucosa caused by abnormal bacterial proliferation in the large intestine [31].

### 3.3. Metabolomics in Sport Medicine

Sport medicine represents one of the most explored topics by equine studies. Unfortunately, it is difficult to draw conclusions that can represent univocal and generalizable results due to the different samples used (serum, urine, saliva), discipline (endurance, trot, race), and study design (different exercise, age, attitude) (Table 3).

Variations in the metabolites in the plasma samples from endurance horses are mainly represented by increases in energetic metabolites such as lactate, pyruvate, creatine, creatinine, and some amino acids [35,36,37]. An increase in lactate and pyruvate after exercise has also been shown in Standardbred and Thoroughbred horses, reflecting the activation of the anaerobic pathway during exercise, as occurs in human athletes [37,38,39,45].

It has also been demonstrated that the race distance in endurance horse (90, 120, or 160 km races), as well the experience of participating in a competition, comparing young and experienced horses, can influence metabolite changes in plasma [36,37]. Due to these multiple variables, further studies based on ^1^H-NMR will help to define the metabolomic profile in horses trained for endurance competitions.

Tyrosine appears to be an interesting molecule in connection with exercise. It has been shown to increase in the plasma of endurance horses [35,37] as well as in the saliva of trotter horses [39]. Tyrosine results from the hydroxylation of the essential amino acid phenylalanine, and it is used as a precursor of catecholamines that play an important role in athletic performance during exercise [46]. It is possible to speculate that the rise in tyrosine after exercise could reflect an increased production to face the need for catecholamines during exercise.

Park and colleagues [38] evaluated the metabolomic profile of muscle and plasma samples in Thoroughbred and Jeju horses before and after exercise. These authors found differences between breeds, with elevated amounts of alanine, methionine, and taurine in plasma pre-exercise in Thoroughbreds but high amounts of aspartate, isoleucine, leucine, and lysine in the skeletal muscles of Jeju horses. The authors concluded that these results suggest the low ability to respond to exercise of Jeju horses and the capacity to perform in races of Thoroughbred horses. Although these results are interesting, it should be considered that the number of animals (limited to three Thoroughbred and three Jeju horses) is probably too low to be representative and allow for the drawing of definitive conclusions.

Zhu and colleagues [40] performed ^1^H-NMR analysis on the urine of ten Standardbred horses before and after a training session. Quinic acid decreased after exercise, while 3-indoxylsulfate and threonine starting to increase. The authors underline how, among these molecules, the latter is the only one that could be directly related to muscular activity. Interestingly, they found some differences linked to gender and age and concluded that the intra-individual variability of urine metabolome might be higher than the inter-individual variability, suggesting the collection of repeated samples from the same individuals to obtain reliable results [40]. The influence on urine metabolome of probiotic supplementation in Standardbred horses before and after exercise has also been investigated [41]. Probiotic supplementation was able to reduce the postexercise blood lactate concentration in Standardbred horses in athletic activity, and this positive effect has been supposed to be connected to a switch of energy source in muscles from carbohydrates to short-chain fatty acids (SCFAs).

Changes related to amino acids and energy metabolism were also found by Jang et al. [42] after exercise. However, this study was carried out on three horses, and the results are limited by the low number of animals used.

Mach et al. [43] associated blood metabolome features to alterations in blood transcriptomes and miRNomes. Eleven metabolites linked to energetic metabolism (glucose homeostasis, lipid metabolism, ketone body generation, ATP synthesis, and acetate production) were found to be altered, but the most interesting conclusion of this paper is related to the ability of this integrated metabolomics/transcriptomic approach to reveal a greater inflammatory status pre-race in the disqualified horses compared to the finishers [43]. This kind of approach could therefore have a predictive value on the performance of endurance horses [43].

Finally, a metabolomic analysis on plasma was applied at a Japanese racing testing center to distinguish samples before and after a jockeyed race [44]. The authors concluded that untargeted metabolomics could be employed in a doping test to find drugs that may be unknown or undetectable with routinary methods that search for pre-determined compounds of known molecular size and weight [8,44].

### 3.4. Metabolomics in Reproduction

Metabolomics applied to animal breeding aim to improve the prediction of the breeding values of the animals to cope with the traditional and new objectives of the selection programs [47].

Metabolomic studies in the field of equine reproduction have been carried out on both mares and stallions (Table 4).

An interesting study performed on mares evaluated, using ^1^H-NMR, the follicular fluid (FF) collected by the transvaginal puncture of the ovary, at various physiological stages of development, together with plasma collection and analysis [48]. The intrafollicular contents of alanine and lipoproteins decreased in dominant follicles during growth, whereas the concentrations of progesterone and estradiol increased significantly. After the induction of ovulation, an increase in lipoproteins and progesterone and decrease in estradiol were observed. The authors concluded that these variations, together with those of several other compounds (among which glycoconjugates, glucose metabolites, amino acids, and polyamines) in relation to growth and maturation are of crucial importance for follicular maturation and ovulation and, consequently, for oocyte maturation and further fertilization. Furthermore, a strong correlation between the intrafollicular content of alanine and circulating estradiol has been observed [48].

In a more recent study, the same authors integrated these results by noticing that several metabolites, including alanine, are much higher in large (dominant) follicles compared to small (subordinate) follicles [49], in a proportion ranging from 250% to as much as 1200%. The authors outline three potential explanations for these findings. On the one hand, the metabolic activity of the granulosa cells within subordinate follicles may be much lower than the one of the granulosa cells within dominant follicles. On the other hand, the density of granulosa cells may be higher in large follicles. Finally, the differences may be due to a change in the permeability of the blood–follicle barrier.

More recently, Fernández-Hernández and colleagues [50] studied the metabolites of preovulatory FF. Twenty-two metabolites were identified, including nine metabolites that are not normally included in the composition of the media used for the in vitro maturation (IVM) of oocyte, namely acetyl carnitine, carnitine, citrate, creatine, creatine phosphate, fumarate, glucose-1-phosphate, histamine, and lactate [50]. It can consequently be supposed that the currently used media for equine oocyte IVM should be further improved to increase the rate of pregnancy success upon fertilization.

In recent years, ‘omics’ studies, including genomics, proteomics, and metabolomics, have been used to search for novel biomarkers of male fertility [57]. The aim was to identify new molecules that could be used to diagnose infertile subjects with increased sensitivity and specificity [58]. Studies have been performed on seminal plasma in humans [58,59,60] and in different animal species [61,62,63,64]. Three of them [51,52,53] have focused on equine seminal plasma, providing interesting insights into the metabolites contained in this biofluid. A preliminary study performed by Magistrini et al. [51] identified, by ^1^H-NMR, metabolites defined as markers of sex gland secretions. By observing three stallions, the authors first evaluated the repeatability of the measures obtained by NMR. Subsequently, they run a qualitative analysis of the NMR spectra to identify the molecules that can be detected in the sex gland secretions and in the seminal plasma. Finally, they performed a quantitative analysis to evaluate the concentration of the components, revealing that nine molecules were differently expressed by the ampulla and bulbourethral glands [51]. A more recent paper by Bazzano and colleagues [52] applied metabolomic analysis to seminal plasma in light and draft horse breeds. This study highlighted features of the metabolome of seminal plasma that differed in relation to horse breed. Moreover, this study found higher concentrations of metabolites like citrate and lactate in the seminal plasma of stallions with a low first cycle fertility rate, as previously observed in other species, including humans, indicating that these metabolites could be observed as possible biomarkers of fertility also in stallions. The concentration of citrate in seminal plasma also resulted in being negatively correlated with the curvilinear velocity (VCL) and average path velocity (VAP) of sperm in a recent paper by Catalàn and colleagues [53]. These authors applied ^1^H-NMR to the seminal plasma of horses and donkeys, showing that the qualitative profile remains consistent across these two species, with 28 common metabolites identified. However, they observed variations in the concentration levels of 18 out of these 28 metabolites. The authors also found that myo-inositol, isoleucine, and valine negatively correlated with progressive motility in donkeys, in agreement with the findings reported in horses by Bazzano et al. [52].

Equine oviductal fluid (OF), at the pre- (PRE) and immediate post-ovulatory (PST) stages, has also been analyzed by ^1^H NMR [54]. The metabolites with the highest concentrations in the OF samples were lactate, myoinositol, creatine, alanine, and carnitine, while fumarate and glycine showed higher concentrations in the PST samples.

Beachler, T. M. et al. collected fetal fluids by ultrasound-guided transabdominal puncture between 270 and 295 days of gestation from 12 pregnant mares [55]. Allantoic fluid resulted in containing betaine, creatine, creatinine, citrate, histidine, nitrophenol, tryptophan, and p-methylhistidine in concentrations higher than amniotic fluid. On the other hand, the amniotic fluid contained increased concentrations of lactate.

More recently, the same authors investigated the plasmatic metabolome of pony mares before and after placentitis induction via *Streptococcus equi* subsp. *zooepidemicus* inoculation in late pregnancy. They noticed a two-phases change in metabolites [56]. Immediately after induction, the concentration of metabolites involved in energy, nitrogen, hydrogen, and oxygen metabolism increased. This phase was followed by a decrease in the concentration of metabolites involved in energy and nitrogen metabolism 4 days after inoculation, when the ultrasonographic diagnosis of placentitis was made. The authors concluded that these changes may represent a diagnostic target for the confirmation of infectious disease in high-risk mares [56].

### 3.5. Metabolomics in Nutrition

Different from other large animal species [65,66,67], few studies have been performed on the effects of nutrition on metabolomic status in horses using ^1^H-NMR [68,69] (Table 5).

These studies have been performed using different matrices (such as serum, urine, feces, and milk) and study designs, hindering an objective comparison of the results.

Leng et al. [68] performed metabolomic analyses on urine and feces collected monthly from 20 ponies over a 13-month period, after dividing them into two groups fed either hay or haylage.

These authors found that urine’s concentration of hippurate and ethyl-glucoside differed between hay-maintained and haylage-maintained ponies, suggesting that forage choice can affect the metabolism of the bacteria that reside within the gut of horses.

The serum of nine Mangalarga Marchador horses has been evaluated before, during, and after a 5-month hypercaloric diet [69]. Choline was found to be related to the diet, so the authors suggested that this molecule could be considered as an ^1^H-NMR potential marker of equine metabolic syndrome (EMS). The increase in high-density lipoproteins (HDL) in horses fed hypercaloric diet also confirms this hypothesis, since HDL is strongly related to insulin resistance, which is the basis of the EMS pathological mechanism [71].

Eighteen metabolites resulted in being different between colostrum and milk collected from female donkeys in the first 45 days of lactation and analyzed by ^1^H-NMR [70]. The same study highlighted the influence of a 30-day probiotic supplementation on donkey milk composition. Twelves metabolites resulted in being significantly different between the supplemented and control group, with higher levels of lactose, O-phosphocholine, and 4-pyridoxate and a lower content of glucose-1-P, caprylate, isovalerate, butyrate, 4-hydroxyphenylacetate, and 2-oxoisocaproate found in the supplemented animals.

Despite the variety of approaches among the above-listed studies, it is possible to state that diet can modulate the metabolome of biological fluids and secretions in equine species.

## 4. Discussion and Conclusions

The application of ^1^H-NMR metabolomics in the field of equine medicine is attracting interest from researchers worldwide. Relatively few papers have been published to date, and the differences in the methods and study designs make it difficult to extrapolate uniquely interpretable data and draw conclusions.

Increasing the quantity of research in this field could allow for a greater collection of data and the possibility of defining new reliable markers to be used in clinical practice, which is not definitively possible at the present time. A Further limitation to the use of NMR is the cost, although the standardization and automatization of the procedures might reduce them to levels of more widespread analytical platforms.

Therefore, this paper may represent a starting point to speed-up the research of some biomarkers. Further interesting evidence is represented by the possibility of coupling metabolomics with other -omics methodologies for diagnostic biomarker discovery from the perspective of a systems biology approach [8].

## Figures and Tables

**Figure 1 metabolites-14-00269-f001:**
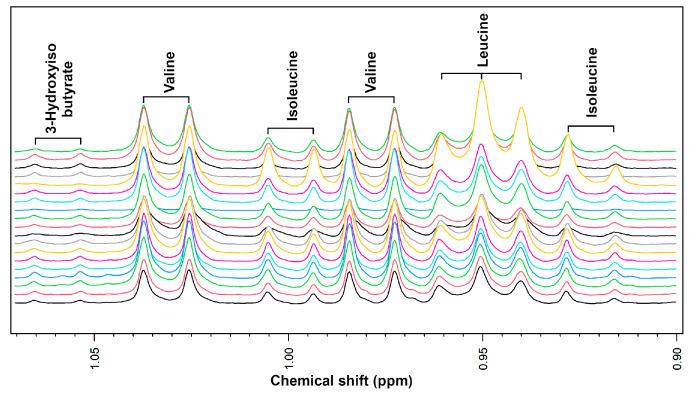
Portion of ^1^H-NMR spectra registered on the synovial fluid of horses’ joints, superimposed in white-washed mode [4].

**Figure 2 metabolites-14-00269-f002:**
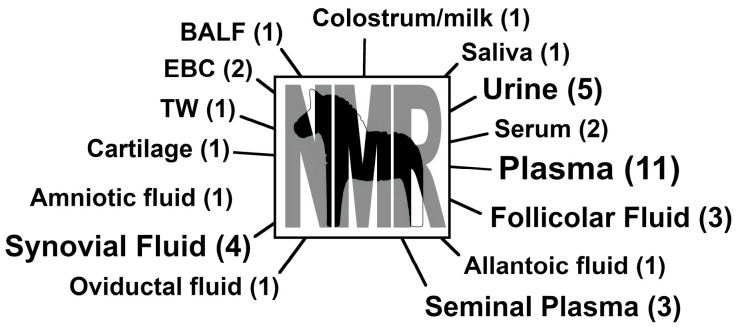
Biological fluids and secretions investigated in equine through a metabolomic approach by ^1^H-NMR. BALF: bronchoalveolar lavage fluid. EBC: exhaled breath condensate. TW: tracheal wash. Numbers of papers investigating each matrix are reported in brackets.

**Table 1 metabolites-14-00269-t001:** Summary of metabolomic studies based on ^1^H-NMR performed on respiratory apparatus. BALF: bronchoalveolar lavage fluid. EBC: exhaled Breath condensate. TW: tracheal wash.

Respiratory System			
Study	Type of Sample	Study Population	Summary of Main Results
[22]	BALF EBC	6 horses with severe asthma 6 healthy horses	Increase in BALF of asthmatic horses: formate, isopropanol. Decrease in BALF of asthmatic horses: myo-inositol, glycerol. Increase in EBC of asthmatic horses: ethanol, methanol.
[23]	TW EBC	6 horses with asthma 6 healthy horses	Increase in TW of asthmatic horses: histamine, glutamate, valine, leucine, isoleucine. Decrease in TW of asthmatic horses: ascorbate, O-phosphocholine, methylamine, dimethylamine, propylene glycol. No differences between groups for EBC.

**Table 2 metabolites-14-00269-t002:** Summary of metabolomic studies based on ^1^H-NMR performed in equine musculoskeletal system. LDLs: low-density lipoproteins.

Musculoskeletal System			
Study	Type of Sample	Study Population	Summary of Main Results
[4]	Synovial Fluid	11 horses with osteoarthritis 8 healthy	Increased in sick horses: 1,3-dihydroxyacetone, 2-hydroxyisobutyrate, creatinine. Decreased in sick horses: 2-hydroxyisovalerate, 3-hydroxybutyrate, 3-methyl-2-oxovalerate, arginine, asparagine, glutamine, glycine, methionine, phenylalanine, tryptophan, tyrosine.
[29]	Synovial fluid	11 horses with osteoarthritis 8 healthy	Increased in affected horse: alanine, acetate, N-acetyle, pyruvate, citrate, creatine, creatinine, choline, glycerol, lactate, glucose.
[30]	Plasma	5 horses before and after laminitis induction using oligofructose	Increased after laminitis induction: oligofructose, lactate, 3-hydroxy-butyrate, glycine, alanine, glutamine. Decreased after laminitis induction: acetate, citric acid, LDLs.
[31]	Synovial Fluid	7 horses with septic joint disease 4 horses with nonseptic joint disease	Higher levels in septic: glycylproline. Higher levels in nonseptic: acetate, alanine, citrate, creatine phosphate, creatinine, glucose, glutamate, glutamine, glycine, phenylalanine, pyruvate, valine.
[32]	Synovial fluid	15 horses affected by palmar osteochondral disease 14 healthy	No statistically significant differences but reduced concentration of glucose and lactate in sick horses.

**Table 3 metabolites-14-00269-t003:** Summary of metabolomic studies based on ^1^H-NMR performed in equine exercise. * Number of horses involved in the study is missing.

Exercise			
Study	Type of Sample	Study Population	Summary of Main Results
[35]	Plasma	29 horses before and after 160 km endurance race	Increased after exercise: lactate, creatine, urea, several amino acids such as valine, leucine, tyrosine. Decreased after exercise: fatty acid chains of lipids, glucosamine.
[36]	Plasma	40 horses before and after 90, 120, and 160 km endurance races	Beta hydroxybutyrate, lactate, acetate, acetoacetate, glutamate, glutamine, creatine, urea, and some other metabolites resulted to be different in function of race distance.
[37]	Plasma	Horses (young and experienced) before and after endurance race	Increased after exercise: -hydroxybutyrate, glycerol, choline, lactate, fumarate, creatine, creatinine, phenylalanine, tyrosine, glutamate, 2-hydroxy-3-methylvalerate. Decreased after exercise: Glucose. Some metabolites resulted to be different among young vs. experienced.
[38]	Muscle biopsy and plasma	3 Thoroughbred and 3 Jeju horses before and after exercise	Higher concentrations of aspartate, leucine, isoleucine, and lysine in the skeletal muscle of Jeju horses than in Thoroughbred horses. Thoroughbred horses had higher levels of alanine and methionine before exercise, whereas postexercise, lysine levels were increased.
[39]	Serum Saliva	12 Standardbred horses before/after full speed exercise	SERUM: Increased after exercise: alanine, citrate, fumarate, glycerol, lactate, leucine, pyruvate, succinate. Decreased after exercise: 3-hydroxybutyrate, 2-hydroxyisobutyrate, acetoacetate, acetone, asparagine, aspartate, creatine, dimethyl sulfone, dimethylglycine, glutamine, histidine, mannose, methanol, myo-inositol, proline, threonine, trimethylamine, valine. SALIVA: Increased after exercise: creatine, ornithine, phenylalanine, sarcosine, tyrosine. Decreased after exercise: 4-aminobutyrate, betaine, fumarate, galactose, malate, malonate, methanol, pyruvate, succinate.
[40]	Urine	10 trotter horse before and after training section	Increased after exercise: 3-indoxylsulfate, threonine. Decreased after exercise: quinic acid.
[41]	Urine	10 Standardbred horses before and after exercise/probiotic supplementation	Increased after exercise because of probiotic supplementation: dimethyl sulfone, pantothenate, taurine. Decreased after exercise because of probiotic supplementation: 2-hydroxyisovalerate, *trans*-aconitate, citrate, *P*-cresol sulfate, glycine.
[42]	Plasma, muscle, urine	3 Thoroughbred after 30 min exercise bout	Alteration of 35 metabolites related to amino acid and energy metabolism.
[43]	Plasma	10 Arabian horses before and after a 160 km endurance competition	11 metabolites linked to energetic metabolism resulted in being different.
[44]	Plasma	30 samples before and after jockeyed race *	Levels of inosine, xanthosine, uric acid, and allantoin, which are induced by extensive exercise, were significantly increased after racing activity in comparison with resting.

**Table 4 metabolites-14-00269-t004:** Summary of metabolomic studies based on ^1^H-NMR performed in equine reproduction. * Number of horses involved in the study is missing.

Study	Type of Sample	Study Population	Summary of Main Results
[48]	Follicular fluid at various physiological stages of follicle development plasma	Pony mares *	The intrafollicular contents of alanine and lipoproteins decreased in dominant follicles during growth. Strong correlation between the intrafollicular content of alanine and circulating estradiol.
[49]	Follicular fluid from large and small follicles	5 samples from large and 5 samples from small follicles of cow, pigs, and mares	Several differences between large and small follicles for equine, including amino acids, creatine, trimethylamine, and glucose, that resulted in being higher in large follicles. Great differences between cow and mares, indicating species-specific differences in follicular metabolism.
[50]	Preovulatory follicular fluid	6 mares	A total of 9 of the 22 metabolites identified are not currently included in the most commonly used media for equine in vitro maturation of oocytes.
[51]	Seminal Plasma	3 stallions	9 metabolites from ampulla and bulbourethral glands.
[52]	Seminal Plasma	6 American Quarter Horse (AQH) and 6 Italian Draft Horse (IDH) stallions	Higher in IDH compared to AQH: 2-hydroxyisobutyrate, 3-hydroxybutyrate, cystine, citrate, glucose, fumarate, hippurate sarcosine, and tyrosine. Lower in IDH compared to AQH: isopropanol, isovalerate.
[53]	Seminal Plasma	18 donkeys and 18 horses	18 metabolites (amino acids, amino acid derivates. and alcohols) resulted in being different between horses and donkeys.
[54]	Oviductal fluid	5 samples per- and post-ovulatory	A total of 18 metabolites were identified with the highest concentrations for lactate, myoinositol, creatine, alanine, and carnitine. Fumarate and glycine were higher in post-ovulatory samples.
[55]	Fetal fluids	7 mares between 270 and 295 days of gestation	Allantoic fluid contained a higher concentration of betaine, creatine, creatinine, citrate, histidine, nitrophenol, tryptophan, and methylhistidine and a lower concentration of lactate compared with amniotic fluid.
[56]	Plasma	10 mares before and after placentitis induction. Five control mares	Four hours post-inoculation, a significant increase was detected in alanine, phenylalanine, histidine, pyruvate, citrate, glucose, creatine, glycolate, hippurate, lactate, and 3-hydroxyisobutyrate. On day 4, a significant reduction in alanine, phenylalanine, histidine, tyrosine, pyruvate, citrate, glycolate, lactate, and dimethylsulfone was seen in infected mares compared with the controls.

**Table 5 metabolites-14-00269-t005:** Summary of metabolomic studies based on ^1^H-NMR focusing on horses’ nutrition.

Study	Type of Sample	Study Population	Summary of Main Results
[68]	Urine Feces	20 ponies divided into two groups: hay vs. haylage feeding	The urinary excretion of hippurate was greater in the hay-fed ponies, while ethyl-glucoside excretion was higher in the haylage-fed ponies.
[69]	Serum	Nine Mangalarga Marchador horses before, during, and after a 5-month hypercaloric diet	Choline resulted to be higher in horse after receiving hypercaloric diet.
[70]	Colostrum and milk	20 jennies before and after probiotics supplementation	Higher concentration in milk after probiotics supplementation: phenylglycine, glucose-1-phosphate, glucose, dimethylamine, 2-oxoisocaproate, glutamine, butyrate, isovalerate, caprylate. Lower concentration in milk after probiotics supplementation: 4-pyridoxate, lactose, O-phosphocholine, ethanol.

## Data Availability

No new data were created in this study. Data sharing is not applicable to this article.

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
