# Peer review of "Nuclear Magnetic Resonance (NMR) Metabolomics: Current Applications in Equine Health Assessment"

_metabolites, 2024, doi:10.3390/metabo14050269_

Round 1

Reviewer 1 Report

Comments and Suggestions for Authors

Dear authors,

thank you for submitting an interesting review of the potential of NMR metabolomics in equine veterinary medicine.

The strength of the manuscript lies in the topic-specific summaries, which do not only provide an overview of the results of each publication, but also put these results in a metabolic context and mentions the potentially involved pathways. However, the manuscript as a whole requires extensive language editing, some rearrangements, and warrant some elaborations in the discussion.

Regarding the structure, it is confusing that (1) the "materials and methods" heading contains a review of the different types of sample matrices and study designs (i.e., a review of the materials and methods) and not the methods used in the review (i.e., how was literature search performed?), and that (2) the sub-headings do not relate to "Materials and methods" as they are topic-specific summaries. I would suggest:

- Adding a "methods" section describing the search strategy (engine, keywords, time frame, date the search was performed, etc)

- Promoting the sub-headings to headings or putting them into a "results" section

- Splitting up the two tables into the corresponding sections (per organ system or condition)

- Adding an overall discussion (see below)

In terms of content, the authors mention infectious diseases in the introduction but no corresponding studies in the results. A short search through google scholar only yielded very few eligible publications (10.1371/journal.pone.0099752, 10.3390/v14071401) but I expect more results if the search is targeted towards specific diseases. While narrative reviews do not require specification of the search methodology, providing a table with all identified publications raises the issue of the search strategy, since the results are not exhaustive. Nevertheless, it would be fine not to include infectious diseases or exclude some results if this is stated in an appropriate section of the manuscript.

Moreover, while the authors provide a good summary of the different studies, the general question of the use of NMR as opposed to other analytical methods is not addressed. In view of the title of the manuscript, I would expect the following questions to be answered in a general section of the discussion:

- Has NMR proven superior to other methods in the equine field or are there specific questions where this method is superior to others?

- Do the results from NMR studies agree with other methods and/or were they confirmed with separate analytical platforms?

- What is the future of NMR in horses? Will it eventually be used in mainstream diagnostics or are there any obstacles to widespread use (standardization, costs, etc.)? Will it be confined to scientific purposes?

In addition, the abstract should summarize the main results/conclusions of the review and not only describe its aims.

Finally, the following inaccuracies should be corrected:

- l. 38-43: the main difference between targeted and untargeted approach is the prior selection and validation of metabolites for the targeted approach, while the untargeted approach is less restricted and will generally use a reference database for metabolite identification but may also describe unidentified metabolites. Moreover, targeted approaches can more easily provide quantitative results. The present description suggests that targeted approaches are inferior as they are "limited" and would not be used to identify biomarkers or distinguish between case and control groups, which is incorrect.

- l. 44-47: a more general statement regarding the coupling of MS would be indicated, as there are many different techniques (e.g., ...-MS/MS, HPLC, FIA, CE, etc.)

- l. 65: the biological fluids were all examined ex-vivo, not in-vivo

Comments on the Quality of English Language

The manuscript requires extensive language editing. Comprehensive corrections were not possible due to time constraints, so the following list of major issues only goes as far as table 1:

- l. 48: is "3" an improperly formatted reference or am I missing the footnotes?

- l. 57: "metabolomic" → "metabolomics"

- l. 59: "contest" → "context"

- l. 61-63: redundant with previous paragraph

- l. 64-67: convoluted sentence, poorly formulated

- l. 73: inconsistent formatting of the table headings

Author Response

Dear Reviewer, authors of the present paper would sincerely thank you for the appreciation of the work done and for the insights to improve the quality of the paper. We have carefully taken into account your recommendations, which has been promptly addressed.

Reviewer 2 Report

Comments and Suggestions for Authors

This review comprehensively explores the applications of 1H-NMR metabolomics in equine health assessment, covering current knowledge, research gaps, methodological approaches, potential clinical uses, and integration with other -omics methodologies. It contributes significantly to expanding knowledge and guiding future research in the field. The discussion on the diagnostic and therapeutic potentials of metabolomics in equine medicine has the potential to inspire new research directions and inform clinical practice. Additionally, the review's examination of combining metabolomics with other -omics approaches for biomarker discovery highlights the importance of a multidisciplinary approach in generating novel insights. However, the absence of figures in the manuscript limits the visual representation of key concepts and data, which could have improved reader understanding and engagement. Including figures in future revisions would enhance the clarity and impact of the manuscript.

Comments on the Quality of English Language

The manuscript could benefit from more consistent use of terminology and clearer transitions between ideas to improve overall coherence.

Author Response

(The authors gave the same response as above.)

Round 2

Reviewer 1 Report

Comments and Suggestions for Authors

Dear Authors,

Thank you for resubmitting a second version of your manuscript. Overall, I find the manuscript to be much improved from the previous version.
The review thoroughly summarises the results of many equine NMR studies. One remaining issue is the discrepancy between the stated and actual scope of the review. The title implies that the usefulness and future perspectives of the method are assessed, and the discussion argues that the paper could be a starting point for future research. However, the manuscript only presents results from NMR studies and at no point does it address the usefulness or limitations of the method (which would imply a comparison with results obtained using other methods). I suggest that the authors either reformulate the title (to be consistent with the summary of NMR studies in horses) and refrain from unwarranted conclusions, or expand the discussion to include the points suggested in my previous review:
1. Has NMR been shown to be superior to other methods in the equine field, or are there specific questions where this method is superior to others?
2. Are the results of NMR studies consistent with other methods and/or have they been confirmed using separate analytical platforms?
3. What is the future of NMR in horses? Will it eventually be used in mainstream diagnostics or are there barriers to widespread use (standardisation, cost, etc.)? Will it be limited to scientific purposes?
... as these should be answered by a review aimed at providing "an overview of [the] usefulness and future perspective [of NMR] in equine health".
I recognise that these questions were touched upon in part in the introduction and mentioned very briefly in the discussion, but this is not sufficient to justify the current title of the review.

Comments on the Quality of English Language

The following typos/formulations should be corrected:
l. 40 define → defines
l. 66 delete "with remarkable ease" (it is not the reproducibility that is facilitated with remarkable ease, the sense of the sentence is unaltered)
l. 191 "which resulted decreased in some cases [4] and increased in others" please clarify the sentence
l. 105 "this is the only study on horses’ plasma laminitis-related" please clarify the sentence
l. 244 "Quinic acid resulted to be decreased after exercise while 3-indoxylsulfate and threonine resulted increased." → remove "resulted to be" and "resulted".
l. 419 "This paper could represent therefore a clarifying starting point for future studies setup to reinforce the trust on biomarkers so far evidenced." → Therefore, this paper could represent a starting point to strengthen the evidence on some biomarkers.

Author Response

Dear Reviewer, the authors would thank you for the time and efforts spent in reviewing the manuscript and for the points raised, which allow to improve the overall quality of the paper. We have carefully taken into account your suggestions and addressed them throughout thew text, using the track change mode.
